# Long-Term Changes in Sleep Disordered Breathing in Renal Transplant Patients: Relevance of the BMI

**DOI:** 10.3390/jcm9061739

**Published:** 2020-06-04

**Authors:** Francesca Mallamaci, Rocco Tripepi, Graziella D’Arrigo, Gaetana Porto, Maria Carmela Versace, Carmela Marino, Maria Cristina Sanguedolce, Giovanni Tripepi, Carmine Zoccali

**Affiliations:** 1CNR-IFC Clinical Epidemiology of Renal Diseases and Hypertension Unit, Center of Clinical Physiology, National Research Council of Italy, Reggio Cal., c/o Ospedali Riuniti, 89124 Reggio Calabaria, Italy; francesca.mallamaci@libero.it (F.M.); rtripepi@ifc.cnr.it (R.T.); g.darrigostat@tin.it (G.D.); tania.porto25@hotmail.it (G.P.); permarica@gmail.com (M.C.V.); cmarino@ifc.cnr.it (C.M.); mcristinas@libero.it (M.C.S.); gtripepi@ifc.cnr.it (G.T.); 2Department of Medicine, Division of Nephrology and Transplantation, Ospedali Riuniti, 89124 Reggio Calabria, Italy

**Keywords:** sleep apnea, renal transplantation, Body Mass Index (BMI), Chronic Kidney Disease (CKD), cardiovascular risk

## Abstract

Sleep disordered breathing (SDB), as defined by the Apnea Hypopnea Index (AHI), is a highly prevalent disturbance in end stage kidney disease. SDB improves early on after renal transplantation but long-term changes in AHI in these patients have not been studied. We studied the long-term changes in AHI in a series of 221 renal transplant patients (mean age: 47 ± 12 years; 70% males) over a median follow up of 35 months. Data analysis was made by the generalized estimating equations method (GEE). On longitudinal observation, the median AHI rose from 1.8 (Interquartile range: 0.6–5.0) to 2.9 (IQR: 1.0–6.6) and to 3.6 (IQR: 1.7–10.4) at the second and third visit, respectively (*p* = 0.009 by the GEE model and the proportion of patients with moderate to severe SDB rose from 8% to 20%. Longitudinal changes in minimum oxygen saturation (minSaO_2_) mirrored those in the AHI. In adjusted analyses, repeated measurements of BMI (*p* < 0.009) emerged as the strongest independent longitudinal correlate of AHI and MinSaO_2_. The AHI worsens over time in renal transplant patients and longitudinal changes of this biomarker are directly related to simultaneous changes in BMI. Overweight/obesity, a potentially modifiable risk factor, is an important factor underlying the risk of SDB in this population.

## 1. Introduction

Sleep disordered breathing (SDB) is a complex disorder which impacts on neurocognitive function, metabolism, cardiovascular health and quality of life and significantly increases the risk of cardiovascular events in the general population [1]. In over 3 million US veterans, incident SDB is associated with higher mortality, incident coronary heart disease, stroke, chronic kidney disease (CKD), and faster CKD progression [2]. SDB is increasingly common when CKD progresses to more severe stages [3] and associates with mortality in the pre-dialysis CKD population [4]. In patients maintained on chronic dialysis, the prevalence of this disturbance attains 56% [5] and entails a high risk of cardiovascular mortality [6,7].

In dialysis patients, SDB is a multifactorial complication which in part depends on compromised upper airway stability secondary to rostral fluid shift overnight [8,9], chronic metabolic acidosis [10], altered central and peripheral chemosensitivity, and probably on the accumulation of still uncharacterized uremic toxins [11] that reduce airway muscle tone. Kidney transplantation restores renal function, corrects fluid overload, metabolic alterations and uremic toxicity and may therefore reverse SDB. Accordingly, in a case–control study the prevalence of SDB among renal transplant patients was similar to that in the general population [12]. Furthermore, with the exception of the study by Tandukar et al., [13] most studies that compared SDB before and after transplantation [12] [14,15,16,17,18] documented that this disturbance improves after kidney grafting.

Risk factors change dramatically after renal transplantation [19,20,21,22]. Among the factors related to SDB, fluid overload [19] and anemia [20] are corrected by renal transplantation early on. On the other hand, after transplantation there is a surge of traditional risk factors [21] of which obesity gradually becomes a prevalent risk factor in this population [22] and fluid overload may re-emerge at later stages [23] in these patients. The long-term dynamics of risk factors for SDB post-transplantation suggests that SDB, after the initial improvement, may not remain stable over time in these patients. In studies performed so far, sleep studies were only performed in the early phase after transplantation. However, there is still no longitudinal study based on repeated sleep studies investigating the long-term evolution of SDB in this population. In particular, the role of overweight and obesity, a major risk factor for SDB which frequently develops after renal transplantation [22], has never been tested in this population in a study with longitudinal design. The issue is important because SDB is a strong predictor of cardiovascular disease events in the general population [24] and entails a high death risk in patients with CKD not on dialysis [4], a category including renal transplant patients [25].

In this study, we investigated whether the improvement in SDB achieved after transplantation is maintained in the long term. With this scope, we enlarged the previous survey at our center [12] from 163 to 221 transplant patients and performed 404 polygraphic sleep studies over a median follow up of 35 months and herein we describe the long-term changes over time in SDB in the same cohort.

## 2. Experimental Section

The protocol was in conformity with the local ethical guidelines of our institution and with the Declaration of Helsinki, and informed consent was obtained from each participant.

### 2.1. Study Design

This is a longitudinal study (median follow up 35 months) embedded in the clinical practice at our Renal Transplantation center. The study includes a baseline sleep recording session performed at least 4 months after transplantation, i.e., when patients had a stable condition, and two or more additional recording sessions at various time intervals.

### 2.2. Patients

The study cohort was composed of 221 renal transplant patients on follow up at the Nephrology, Dialysis and Transplantation Unit of Reggio Calabria, Italy. The flow chart of this cohort is shown in Figure 1.

### 2.3. Study Procedures

Polygraphic sleep study: All recording sessions were carried out while patients were in a stable condition and without intercurrent clinical problems. Polygraphic recordings were performed with patients sleeping in a quiet single bedroom or in a double room where the room mates were in stable conditions and did not need night care [12]. Bedtime and awakening times were at each subject’s discretion and the start of the sleeping time was declared by the patient, which was checked based on gross movements records and confirmed by a nurse who periodically checked the patient. The study consisted of continuous polygraphic cardiorespiratory recording from surface leads for electrocardiography and from non-invasive sensors for nasal airflow, thoracic and abdominal respiratory effort, oxyhemoglobin level (finger-pulse oximeter) and gross movements records. Until 2009, we used the Pamela Sleep Recorder by MEDATEC, Medigas, Milan, Italy and, as of 2009, the Somté recorder by Compumedics, Victoria, Australia distributed in Italy by MEDIGAS, Assago-Milan, Italy. For nasal airflow, the Pamela recorder used a thermistor, while the Somté recorder uses a flowmeter. The transducers and lead wires permitted normal positional changes during sleep. The polygraphic recording was terminated after final wakening which coincided with the time adopted for the calculation of the total sleep time. Laboratory data: Laboratory data either collected during the same day of the sleep study or during the visit preceding the same recording (in general within 2 weeks) were abstracted from electronic clinical files of our Unit. Serum lipids, glucose, albumin, phosphate, PTH, and hemoglobin were measured by standard methods in the routine clinical laboratory. C-reactive protein (hs-CRP) was measured by a high sensitivity method (Dade Behring, Marburg, Germany). Serum creatinine was measured by an automated technique based on the Jaffe chromogen method (calibrated to the Isotope Dilution Mass Spectrometry standard, IDMS) implemented in an auto-analyzer. The Modification of Diet in Renal Disease (MDRD) equation developed by Levey et al. [26] estimated the glomerular filtration rate (eGFR).

### 2.4. Assessment of Sleep Disordered Breathing

Records were then scored for sleep, breathing, oxygenation, and movements in 30-second periods. An abnormal breathing event during sleep was defined as a complete cessation of air flow lasting 10 s or more (apnea) or a discernible reduction in respiratory airflow accompanied by an oxygen saturation drop of >4% (hypopnea) [26]. The average number of episodes of apnea and hypopnea per hour of sleep (the Apnea Hypopnea Index, AHI) was calculated as the summary measure of sleep-disordered breathing. At all study visits, for the categorical analysis, we applied the AHI cut off points adopted in the Wisconsin study [27]: 0–< 5 apnea-hypopnea episodes (normal sleep pattern), >5–< 15 episodes (mild SDB), >15 < 30 episodes (moderate SDB) and ≥30 episodes (severe SDB). Nocturnal hypoxemia was measured by considering the minimal and average O_2_ saturation as well as the number of O_2_ desaturation episodes (Oxygen Desaturation Index, ODI) during nighttime. A fall in oxygen saturation was considered significant if the oxygen desaturation during the episode was ≥4% of the surrounding values. Nocturnal apneas were further classified according to recommendations by the American Academy of Sleep Medicine Task Force [28]. Briefly, an apnea episode was classified as ‘central’ when it was associated to a fall in inspiratory muscles activity following an exhalation and as ‘obstructive’ when inspiratory muscle activity was present without airflow. An observer that was kept blind to the scope of the study analyzed all recordings.

### 2.5. Statistical Analysis

Data are summarized as mean and standard deviation, median and interquartile range, or as percent frequency as appropriate. Comparisons among AHI categories were performed by One Way ANOVA (for normally distributed continuous variables), Kruskall–Wallis Test (for non-normally distributed continuous variables), or Chi-Square Test, as appropriate. The relationship between non-normally distributed variables was investigated by Spearman Rank Correlation coefficient (rho) and *p* values. The study had no missing data for the variables included in the multivariable models.

The relationship between repeated polygraphic sleep measures (namely: AHI, ODI, minimum O_2_ and average nocturnal O_2_ saturation) and time was investigated by generalized estimating equations (GEE) [29]. The correlates of polygraphic sleep data, which significantly changed over time, were analyzed by multiple GEE models including all factors which differed among AHI groups (see Table 1) with *p* ≤ 0.10. In these models, BMI and eGFR were introduced as repeated measurements. Based on the distribution of the key dependent variables (namely: AHI and minimal nocturnal O_2_ saturation), GEE models were fitted by adopting a Tweedie distribution [30] with a log10 based link function. In GEE models, data are expressed as regression coefficients, 95% CI and *p* value. Data analysis was performed by SPSS^®^ 24.0 (IBM Corporation, Armonk, NY, USA) for Windows^®^.3.

## 3. Results

The total transplant population on follow up at our institution during the enrolment years (March 2004–June 2015) counted 340 patients. Among these, 47 patients refused to participate in the study and the remaining 72 did not complete polygraphic recordings for technical reasons or polygraphic recordings could not be arranged for logistic reasons (Figure 1). Thus, the final study population was composed of 221 renal transplant patients.

### 3.1. Description of the Study Cohort

As shown in Table 1, the mean age was 47 ± 12 years and 70% were males. Nine percent were diabetics. The causes of CKD were: Glomerulonephritis in 82 cases (37.%), diabetic kidney disease in 18 cases (8%), cystic kidney diseases in 20 cases (9%), interstitial nephropathy in 13 cases (6%), congenital anomalies of the kidney and urinary tract (CAKUT) in 10 cases (4%), hypertensive nephrosclerosis in 2 cases (1%), hereditary nephropathies in 2 cases (1%), acute kidney injury (AKI) in 2 cases (1%), and other causes/unknown in the remaining 72 cases (33%). Most patients received the organ from cadaveric donors (88%) and a minority from living donors (12%). The majority (71%) were on triple immunosuppressive therapy (three drug combinations among cyclosporine, steroids, azathioprine, tacrolimus, sirolimus and mycophenolate) and the remaining patients were on double therapy (28%) or on mono-therapy (1%) with these agents.

Twenty-four patients had experienced cardiovascular events. In detail, eighteen had suffered from one event (myocardial infarction in three, stroke in four, transient ischemic attack in three, angina in two, arrhythmia in four, and peripheral vascular disease in two patients) and the other six had had two or more of these events. Ten percent of the patients were active smokers and 41% past smokers. One hundred and ninety-five patients were on anti-hypertensive therapy. Sixty-seven patients were on monotherapy with calcium channel blockers (*n* =12), beta-blockers (*n* = 25), ACE inhibitors/angiotensin II receptor blockers (ARB) *n* = 24), sympatholytic or vasodilatatory agents (*n* = 5) and diuretics (*n* = 1). One hundred and twenty-eight patients were on multiple therapies with various combinations of these drugs. Ten percent of the patients were being treated with erythropoietin stimulating agents (ESA) and 42% were on treatment with statins. Estimated glomerular filtration rate (eGFR) was on average 56.1 ± 20.5 mL/min/1.73 m^2^ (range 7.2–135.9 mL/min/1.73 m^2^). Office BP was on average 132 ± 16/78 ± 10 mmHg (Table 1).

### 3.2. Baseline Polygraphic Sleep Data

At baseline, the median value of theAHI was 1.8 episodes/h (interquartile range: 0.6–5.0 episodes/h). One hundred and sixty-six patients (75%) had an AHI <5; 37 patients (17%) had an AHI ranging from 5 to <15 and the remaining 18 patients (8%) had an AHI >15. This prevalence of SDB is substantially similar to that observed in the general population matched for age, gender and BMI [29] (Figure 2).

The AHI was directly related to age (Spearman rho = 0.24, *p* < 0.001) and BMI (rho = 0.28, *p* < 0.001) (Table 1). The eGFR was similar across the three AHI strata (*p* = 0.10). As expected, the ODI paralleled the AHI while the average O_2_ saturation and the median minimum O_2_ saturation gradually reduced across AHI categories denoting SDB of increasing severity (Table 1). In Appendix A, drug treatments are reported across the same categories.

AHI was directly related to the use of statins, calcium antagonists, sympatholytic agents/vasodilators and the total number of anti-hypertensive drugs and inversely to the use of ESA and steroids. However, these associations were no longer significant after simple data adjustment for the BMI (*p* ranging from 0.09 to 0.89).

### 3.3. Longitudinal Polygraphic Sleep Data

Over a median follow up of 35 months (whole range 4 to 110 months; interquartile range 24–46 months), 404 polygraphic recordings were performed. In detail, 82 patients were only studied at baseline, 139 patients had at least two polygraphic studies and 44 had three studies. Face to face comparison of patients only studied at baseline (*n* = 82) did not differ from the remaining patients (*n* = 139) as to age (47 ± 11 vs. 47 ± 12 years, *p* = 0.74), 24 h systolic BP (126 ± 14 vs. 124 ± 11 mmHg, *p* = 0.48), BMI (26.3 ± 3.9 vs. 25.6 ± 3.4 k/m^2^, *p* = 0.18), diabetes (9.8% vs. 8.6%, *p* = 0.78), male sex (69.5% vs. 70.5%, *p* = 0.88), and background CV comorbidities (13.4% vs. 9.4%, *p* = 0.36).

The median AHI rose from a baseline median value of 1.8 episodes/h (IQR: 0.6–5.0) to 2.9 episodes/h (IQR: 1.0–6.6) at the second visit (time spanning from baseline, median: 32.2 months, IQR: 32.2–41.0 months) and to 3.6 episodes/h (IQR: 1.7–10.4)) at the third visit (time spanning from baseline, median: 52 months, IQR: 36.8–67.3 months) (*p* = 0.009, calculated by the GEE model). In a sensitivity analysis restricted to the subgroup of patients with three sleep recordings (*n* = 44) the worsening AHI (baseline: 1.3 episodes/h, IQR 0.50–2.3; first visit, 2.0 episodes/h, IQR 0.7–3.3; second visit, 3.6 episodes/h IQR 1.7–8.5) was similar to that observed in the primary analysis (see above). In a categorical analysis, the prevalence of moderate to severe SDB rose markedly across the follow up, from 8% (18/221 patients) at the first visit to 13% at the second visit (18/139 patients) and 20% (9/44 patients) at the last visit (Figure 3).

The time trend of minimal nocturnal O_2_ saturation mirrored the AHI trend (first visit, 87 ± 7%; second visit, 87 ± 7%; third visit, 85 ± 11%, *p* = 0.048 by the GEE model). Neither the ODI (*p* = 0.27), nor the average nocturnal O_2_ saturation (*p* = 0.14) changed significantly over time. To identify risk factors for SDB worsening over time, we built a GEE model of longitudinal changes of the AHI (Table 2) including age, gender, diabetes, and repeated measures of the BMI and eGFR, i.e., the variables that were associated with the AHI at baseline with a *p* value <0.10. In this analysis, age, gender, and repeated measurements of BMI emerged as the strongest correlates of the longitudinal changes of the AHI. In a multivariate GEE model looking at changes in minimal nocturnal O_2_ saturation, age, gender and repeated measurements of BMI were again independent correlates of this indicator of SDB (Table 2).

The recorded change over follow up (Pamela → Somtè, see methods) was associated per se with the change in AHI and minimal O_2_ saturation over follow up but it did not confound the longitudinal relationship of the BMI and other variables with the same outcome measures (Appendix A). Accordingly, in a formal effect-modification analysis testing the interaction term sleep recorder change (Pamela → Somté) and study visit in the same model described in Table 2, this interaction term was largely insignificant both for the AHI ((*p* = 0.62) and the minimal O_2_ saturation (*p* = 0.74). The BMI and AHI evolved in parallel (*p* = 0.002) in patients for whom the BMI increased by >0.5 kg/m^2^ but were dissociated in those for whom the BMI remained stable or declined (*p* = 0.48).

## 4. Discussion

We found that 8% of the renal transplant patients studied, after having achieved clinical stability after kidney grafting, had moderate to severe SDB and an additional 17% had mild SDB. These figures are substantially less than those typically seen in dialysis patients where moderate to severe SDB has a 56% prevalence [5] and are similar to those in age-, sex- and BMI-matched individuals in the general population [12,31]. However, we also found that SDB worsens in these patients over time and, after a median follow up of 3 years, 20% of the renal transplant patients had moderate to severe SDB. Furthermore, a BMI change over time was the sole modifiable risk factor underlying SDB worsening in this population.

Restored renal function after transplantation normalizes several risk factors associated with SDB in dialysis patients. Fluid overload, uremic toxicity and other factors are rapidly corrected after transplantation. However, after the early improvement, fluid overload re-emerges over time and at 5 years post-transplantation the proportion of transplant patients with fluid overload attains 30% [23]. Similarly, body weight gradually increases over time post-transplantation and obesity is recognized as a highly prevalent (30% to 50%) [32,33], high-risk problem [32,33,34] in the renal transplant population. Thus, after the early improvement, renal transplant patients may be exposed to a high risk of SDB recurrence. The initial beneficial effect of renal transplantation on SDB is generally considered to be a stable improvement and perhaps for this reason SDB is not considered as a possible risk factor for adverse cardiovascular outcomes in the transplant population [21,35].

Given the strong link between SDB, death [4], and cardiovascular events [2,6,7] in the CKD population, testing the long-term evolution of SDB in renal transplant patients is a relevant issue for our understanding of the potential risk of SDB for these outcomes in the renal transplant population. With this study, we confirm our previous findings in a larger population [12] that, in an early phase, stable renal transplant patients have no excess risk of SDB as compared to well-matched individuals in the general population. However, we also found a progressive worsening of this alteration after long-term longitudinal observation. Indeed, the prevalence of moderate to severe SDB in the baseline study was only 8% but increased to 20% over a median follow up of 3 years, which is more than double that registered at baseline and in the coeval general population [29]. In a large study with 2921 elderly people in the general population, a 20% prevalence in SDB was only observed in patients between 65 and 72 years of age [36] i.e., in people about 20 years older than the renal transplant patients included in the present study. The recurrence of SDB after transplantation may have clinical implications. Indeed an AHI between 4 and 12, an alteration we found in one out of four male and in one out of 11 female transplant patients in this study, entails a 75% excess risk of death and stroke in patients with suspected sleep apnea [37]. Of note, we found that SDB worsening in transplant patients over time associated with simultaneous changes in the BMI. BMI is a robust causal risk factor for SDB in the general population [38]. A link between the BMI and SDB in transplant patients emerged in a cross-sectional study by Molnar et al., [39] and in a survey by Mallamaci et al. [12]. This longitudinal study is the first to focus on the long-term changes of AHI after transplantation. We enrolled a number of patients (*n* = 221) about 2.8 times higher than the aggregate population (*n* = 80) of previous studies that measured AHI just before and a relatively short time after renal transplantation [13,16,40,41]. Our observations are based on over 400 polygraphic recordings distributed over a median follow up of 3 years. On longitudinal analysis, we found an independent relationship between the BMI and the AHI. Longitudinal studies provide more robust information than cross sectional studies for the assessment of causality [42]. Thus, the present study supports the hypothesis [12,39] that a high BMI is a causal risk factor for SDB in transplant patients.

This study has limitations. One third of the patients were only studied at baseline and only 44 patients had three sleep recording studies. However, the generalized estimating equations model adequately deals with the unequal number of patients at various time points and with the variable time interval between successive recordings [29]. Our population includes only Caucasian patients in a single transplant center. Therefore, our results need to be confirmed by studies in other transplant center enrolling patients of other races and ethnicities. Another limitation is the fact that we used two polygraphic recorders with different airflow sensors (thermistor and airflow pressure detector, respectively). However, the progressive rise of AHI and its relationship with BMI was largely independent of the type of polygraphic sleep recorder. Furthermore, for technical reasons we could not produce an analysis according to apnea type and the time spent with SaO_2_ < 90%. In addition, we did not measure body fluids. Therefore, we could not test the relevance of fluid overload as a long-term risk factor for SDB. Finally, our data are observational in nature. The randomized clinical trial is the standard for testing causal hypotheses. Trials focusing on renal transplant patients would be important to understand the effects of obesity on health in the specific context of this population. Future trials targeting obesity in transplant patients may include polygraphic recordings to test the hypothesis that the longitudinal link between BMI and SDB observed in the present study is causal in nature.

In conclusion, this longitudinal study shows that sleep disordered breathing worsens over time in renal transplant patients. In the context of the present study, the BMI emerged as the sole modifiable risk factor underlying SDB worsening in this population.

## Figures and Tables

**Figure 1 jcm-09-01739-f001:**
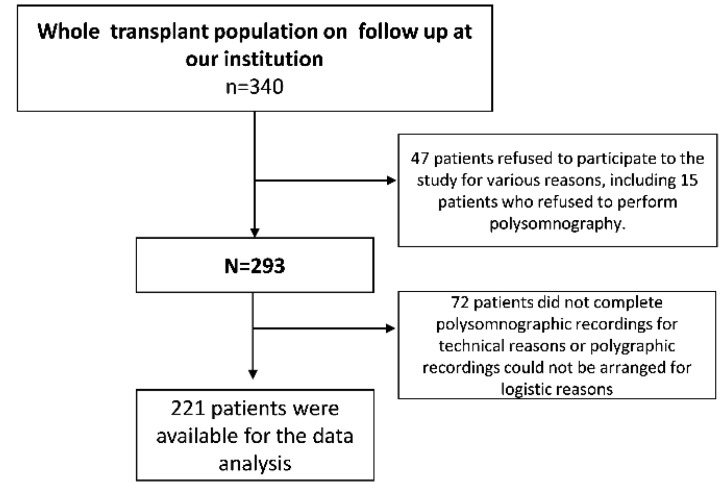
Flow chart of patients included in the study.

**Figure 2 jcm-09-01739-f002:**
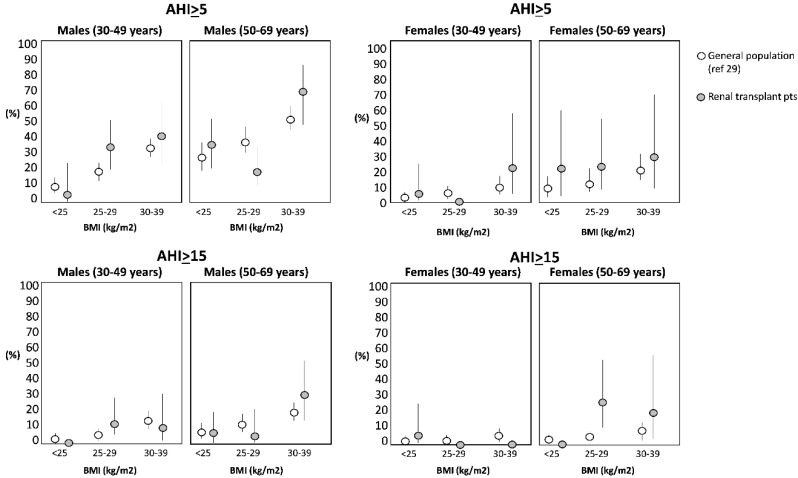
AHI stratified by gender, and BMI in transplant patients in the present study (grey circles) and in individuals in the general population (white circles) described in [29].

**Figure 3 jcm-09-01739-f003:**
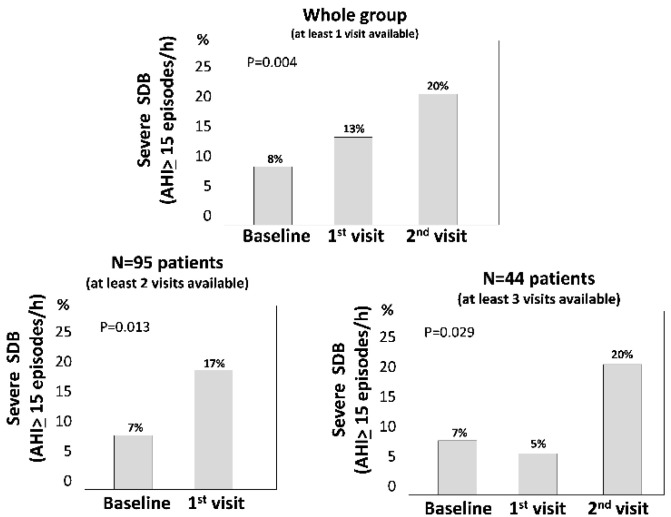
Prevalence of moderate to severe sleep disordered breathing (SDB) across the study in the whole cohort, in patients who had at least 2 visits and in those with at least 3 visits.

**Table 1 jcm-09-01739-t001:** Main demographic, clinical and biochemical baseline characteristics of patients grouped by baseline apnea-hypopnea index. * ANOVA; ^ Chi Square; ^#^ Kruskall–Wallis.

Baseline Values	Baseline Apnea-Hypopnea Index (Episodes/h)				
	Whole Group (*n* = 221)	<5.0 (*n* = 166)	from ≥5 to <15 (*n* = 37)	≥15 (*n* = 18)	*p* (See also Statistical Methods)
Age (years)	47 ± 12	45 ± 12	51 ± 10	53 ± 9	0.001 *
Organ from living donors (%)	12%	11%	19%	6%	0.99 ^
Male sex (%)	70%	66%	89%	72%	0.065 ^
Active smokers (%)	10%	10%	10%	14%	0.70 ^
Past smokers (%)	41%	38%	48%	50%	0.23 ^
Diabetes (%)	9.0%	8%	8%	22.%	0.097 ^
Background CV complications (%)	11%	10%	11%	22%	0.16 ^
Systolic BP (mmHg)	132 ± 16	132 ± 15	134 ± 15	136 ± 23	0.24 *
Diastolic BP (mmHg)	78 ± 10	78 ± 10	78 ± 10	79 ± 9	0.69 *
Sodium (mEq/L)	140.0 ± 4.7	139.9 ± 4.9	140.2 ± 4.9	140.7 ± 2.5	0.51 *
Potassium (mEq/L)	4.2 ± 0.5	4.2 ± 0.5	4.1 ± 0.5	4.4 ± 0.5	0.33 *
Cholesterol (mg/dL)	180 ± 36	180 ± 37	177 ± 33	182 ± 37	0.96 *
HDL cholesterol (mg/dL)	55 ± 16	55 ± 15	55 ± 20	51 ± 11	0.34 *
LDL cholesterol (mg/dL)	100 ± 34	102 ± 36	92 ± 27	104 ± 33	0.64 *
BMI (kg/m^2^)	25.9 ± 3.6	25.4 ± 3.3	27.2 ± 4.0	28.1 ± 4.0	<0.001 *
Hemoglobin (g/dL)	13.0 ± 1.6	12.9 ± 1.7	13.3 ± 1.4	12.6 ± 2.0	0.998 *
Albumin (g/dl)	4.2 ± 0.4	4.2 ± 0.4	4.1 ± 0.4	4.1 ± 0.3	0.18 *
Phosphate (mg/dl)	3.3 ± 0.8	3.3 ± 0.8	3.2 ± 0.7	3.7 ± 0.8	0.16 *
PTH (pg/mL)	67 (43–106)	67 (41–100)	64 (45–191)	70 (49–115)	0.72 ^#^
hs-CRP (mg/L)	1.5 (0.6–3.1)	1.4 (0.6–2.9)	1.2 (0.5–3.4)	2.4 (1.3–7.2)	0.58 ^#^
eGFR-MDRD_186_ (mL/min/1.73 m^2^)	56.1 ± 20	56 ± 21	56 ± 18	58 ± 21	0.10 *
**Polygraphic Sleep Data**					
Apnea-hypopnea index (episodes/h)	1.8 (0.6–4.9)	1.1 (0.5–2.2)	7.5 (5.9–9.9)	28.5 (19.5–51.2)	<0.001 ^#^
ODI (number of O_2_ desaturation episodes/h)	1.30 (0.30–4.45)	0.70 (0.18–2.23)	5.6 (2.5–8.6)	18.6(11.0–43.7)	<0.001 ^#^
Minimum O_2_ saturation, min SaO_2_ (%)	89 (86–92)	90 (88–93)	86.3 (80–88)	80.0 (70–88)	<0.001 ^#^
Average O_2_ saturation, mean SaO_2_ (%)	96 (94–96.)	96 0(94–97)	95 (93–96)	94 (90–96)	<0.001 ^#^

Data are expressed as mean ± SD, median and interquartile range or as percent frequency, as appropriate.

**Table 2 jcm-09-01739-t002:** Adjusted multiple generalized estimating equations (GEE) of Apnea Hypopnea Index (AHI) and minimal nocturnal O_2_ saturation over time. The model includes all univariate correlates of AHI at baseline with *p* < 0.10 (age, gender, diabetes as well as the BMI and estimated glomerular filtration rate (eGFR). BMI and eGFR are tested as repeated measures). Data are regression coefficients, 95% CI and *p* value.

	Apnea Hypopnea Index	Minimal O_2_ Saturation
*Variables (units of increase)*	*Regression coefficients (95% CI)*	*Regression coefficients (95% CI)*
Age (5 years)	1.105 (1.017–1.201), *p* = 0.019	0.996 (0.992–0.999), *p* = 0.02
Male/females	1.725 (1.014–2.935), *p* = 0.044	0.980 (0.963–0.997), *p* = 0.02
Diabetes (yes/no)	1.217 (0.483–3.069), *p* = 0.68	0.960 (0.904–1.025), *p* = 0.25
* BMI (1 kg/m^2^)	1.098 (1.037–1.164), *p* = 0.001	0.990 (0.980–0.996), *p* < 0.001
* eGFR (1 mL/min/1.73 m^2^)	1.001 (0.991–1.011), *p* = 0.90	1.000 (0.999–1.000), *p* = 0.92
Visit (0,1,2)	1.409 (1.170–1.697), *p* < 0.001	0.985 (0.972–0.998), *p* = 0.02

* BMI and eGFR are introduced into the model as repeated measurements.

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
