# Peer review of "Long-Term Changes in Sleep Disordered Breathing in Renal Transplant Patients: Relevance of the BMI"

_jcm, 2020, doi:10.3390/jcm9061739_

Round 1
Reviewer 1 Report
Report entitled „ Long Term Changes in Sleep Disordered Breathing in Renal Transplant Patients” reports on the increase in AHI and in the frequency of moderate to severe SLB in individuals following a kidney transplant.
I have a question regarding the scoring criteria used in the long period of time that patients were recruited. Authors state that discernible reduction in respiratory airflow accompanied by an oxygen saturation drop of >4% was considered as a hypopnea. In recent criteria the saturation drop is >3% or >4%with the presence of arousal.
I recommend using more recent clinical and diagnostic guidelines that 1999 f.e. VK Kapur, DH Auckley, S Chowdhuri, et al. Clinical Practice Guideline: Diagnostic Testing OSA; http://dx.doi.org/10.5664/jcsm.6506.
The study design should be clearly described in the Methods section – the number of PSG examination and time points, at which they were recorded should be presented.
In baseline characteristics, where authors describe the percentage of central/obstructive events in individuals (line 190, page 5) it should be done more clearly in the same way for each of the event type – mixed is described in percentage on individuals and obstructive and central should be done similarly. Further, are how many individuals were diagnosed as OSA and how many as CSA – finding predictive factors for both together is questionable as for both disorders different parameters are considered as risk factors.
In Table 1 parameter described as “Number of O2 desaturation episodes (episodes/h)” should be corrected as desaturation index (ODI) (number of events over time – similarly to AHI).
Figure/table representing longitudinal analysis including separately whole group, a group that has 2 PSG recordings and group were 3 recordings were performed should be provided.
In Figure 3 and related description authors define severe SDB as AHI>15, while earlier describing it as moderate to severe (which is the correct description). It should be corrected as it is misleading. Additionally, while marking the percentage of individuals with subgroups of SBD on this Figure please provide additionally n of patients, as the n is provided as a baseline characteristic.
Were the baseline examinations at the same time before the transplant or what was the period difference between individuals?
I am not clear if different time points of the follow-up PSG examinations were considered as confounding factors in the analysis.
Please edit the manuscript carefully in many places it is missing commas and periods. Correct also for typos.
Author Response
Q1
I have a question regarding the scoring criteria used in the long period of time that patients were recruited. Authors state that discernible reduction in respiratory airflow accompanied by an oxygen saturation drop of >4% was considered as a hypopnea. In recent criteria the saturation drop is >3% or >4%with the presence of arousal.
R: We are aware that more recent criteria accept a O2 saturation of 3% drop as sufficient for the adjudication of apnea episodes. However, our study is a long study started in March 2004 and our protocol was established on the standard, internationally accepted criteria of the time (2004). We could only maintain the same criteria throughout the study to ensure analytical coherence.
Q2
I recommend using more recent clinical and diagnostic guidelines that 1999 f.e. VK Kapur, DH Auckley, S Chowdhuri, et al. Clinical Practice Guideline: Diagnostic Testing OSA; http://dx.doi.org/10.5664/jcsm.6506.
R: we thank the referee for this reference. We believe that our study protocol complies with most recommendations in these recent diagnostic guidelines. However, we feel that we cannot quote these guidelines because our study was started in 2004 and, as commented in the reply to the first question, we had to maintain the criteria of the time when the study was designed .
Q3
The study design should be clearly described in the Methods section – the number of PSG examination and time points, at which they were recorded should be presented.
R: Ours is a study embedded in clinical practice. Our aim was to repeat sleep studies at least twice per patient at an interval of about 2-3 years. Therefore the time was not exactly pre-fixed. In the results we give detailed information on the 404 recordings performed during longitudinal observation (Pg 6, L568-570). In detail, 82 patients were studied only at baseline, 139 patients had at least two polygraphic studies and 44 had three studies. We also specified (Pg 6 L581-586) that the median time spanning from the second visit to baseline was 32.2 months ( Inter Quartile Range: 32.2-41.0 months) and that the median time spanning from the third visit to baseline was 52 months (IQR: 36.8-67.3 months). To deal with the problem of unequally spaced and differently available time points and to include also individuals with just one recording, we adopted the Generalized Estimating Equations approach, a statistical technique which is adequate for the analysis of our data (see new reference 29: Hanley JA, Negassa A, deB Edwardes MD, Forrester JE. Statistical Analysis of Correlated Data Using Generalized Estimating Equations: An Orientation. Am J Epidemiol 2003; 157:364-375).
Q4
In baseline characteristics, where authors describe the percentage of central/obstructive events in individuals (line 190, page 5) it should be done more clearly in the same way for each of the event type – mixed is described in percentage on individuals and obstructive and central should be done similarly. Further, are how many individuals were diagnosed as OSA and how many as CSA – finding predictive factors for both together is questionable as for both disorders different parameters are considered as risk factors.
R: In rechecking data about the type of apnea we realized that the figures about central and mixed apnea at baseline were wrong. Ours is a study started in 2004 and information on apnea types was not systematically transcribed in the study data base. We planned to recheck all polygraphic recordings of our study and to this scope asked to the Journal an extension of the deadline for resubmission. Unfortunately we discovered that recordings from March 2004 to August 2007 (our first survey including 163 patients which we extended to 221 to form the basis of the present longitudinal study) were irretrievable. Even though restricted to the AHI, we feel that our study contains valid, novel data that may contribute to knowledge in the field. We are pretty sure that the vast majority of apneas were obstructive but we cannot provide data to support this statement. If the referees believe that information about apnea types is obligatory, we are sorry to lose the opportunity to publish the paper in the JCM but we can only withdraw our submission.
Q5
In Table 1 parameter described as “Number of O2 desaturation episodes (episodes/h)” should be corrected as desaturation index (ODI) (number of events over time – similarly to AHI).
R: We redefined “number of desaturation episodes” as ODI in Table 1 and maintained this designation throughout the manuscript (Pg3 , L418 and L431; Pg 6, L496; Pg 6 L533)
Q6
Figure/table representing longitudinal analysis including separately whole group, a group that has 2 PSG recordings and group were 3 recordings were performed should be provided.
R: We have now added in Figure 3 the separate analysis in patients with at least 2 and 3 polygraphic recordings
Q7
In Figure 3 and related description authors define severe SDB as AHI>15, while earlier describing it as moderate to severe (which is the correct description). It should be corrected as it is misleading. Additionally, while marking the percentage of individuals with subgroups of SBD on this Figure please provide additionally n of patients, as the n is provided as a baseline characteristic.
R: We corrected the inappropriate labeling “severe” with “moderate to severe” both in the figure and in the text (pg 6 L529-532). The absolute numbers were 18/221 and 18/ 139 at baseline and at the first visit and 9/44 at the second visit (ibidem).
Q8
Were the baseline examinations at the same time before the transplant or what was the period difference between individuals?
R: As specified in the text, (Pg 2, L74-77) all measurements were performed after transplantation, with the baseline measurement made after at least 4 months after transplantation i.e. when the patients had a achieved a stable condition after the intervention.
Q9
I am not clear if different time points of the follow-up PSG examinations were considered as confounding factors in the analysis.
R: The time points were incorporated in the analysis to test whether the longitudinal increase of AHI over time is significant and the study visit remained as a significant predictor of the AHI.
Q10
Please edit the manuscript carefully in many places it is missing commas and periods. Correct also for typos.
R: We have asked our proof reader to thoroughly recheck language of our manuscript.
Reviewer 2 Report
This interesting study investigated if SDB improved or impaired over a median follow-up of 35 months in renal transplant patients. The authors observed that SDB, defined with an AHI index > 5, affected 8% of transplant patients at baseline, whereas increased to 20% at the third PSG. Interestingly, the authors reported that, in addition to advanced age and sex male, BMI was the only other predictor of SDB. Overall, the paper was well written and statistical analysis was accurate. However, I would give authors some suggestions for improving their study.
- Since BMI was the only modifiable predictor of impaired SDB over time, the authors should report the AI for both obstructive and central events. I think that OSAS and not CSAS affected transplant patients over time. Please, confirm or confute this hypothesis.
- Since the use of hypnotic drugs causes the occurrence of apnea/hypopnea events, please collect, analyze and report information on these drugs in the text.
- The authors explored the following polygraphic sleep measures: AHI, number of O2 desaturation episodes per hour, minimum O2 and average nocturnal O2 saturation. Please, add also the time spent with O2 saturation below 90%.
- Figure 2 was irrelevant. Please, delete.
- Page 5, line 193: Delete correlations between AHI and age, BMI.
- Change the title underling the importance of BMI as predictor of SDB in transplant patients.
- Please, include among limitations that more than one third of patients were studied only at baseline and that only 44 patients had three PSG studies.
Author Response
Referee #2
Q1
Since BMI was the only modifiable predictor of impaired SDB over time, the authors should report the AI for both obstructive and central events. I think that OSAS and not CSAS affected transplant patients over time. Please, confirm or confute this hypothesis.
R: As we specified in the reply to Q4 by Referee#1, in rechecking data about the type of apnea we realized that the figures about central and mixed apnea at baseline were wrong. Ours is a study started in 2004 and information on apnea types was not systematically transcribed in the study data base. We planned to recheck all polygraphic recordings of our study and asked to the Journal an extension of the deadline for resubmission. Unfortunately we discovered that recordings from March 2004 to August 2007 (our first survey including 163 patients which we extended to 221 to form the basis of the present longitudinal study) were irretrievable. We are aware that this information is of relevance. Even though restricted to the AHI, we feel that our study contains valid, novel data that may contribute to knowledge in the field. We are pretty sure that the vast majority of apneas were obstructive but we cannot provide data to support this statement. If the referees believe that information about apnea types is obligatory, we are sorry but we can only withdraw our submission.
Q2
Since the use of hypnotic drugs causes the occurrence of apnea/hypopnea events, please collect, analyze and report information on these drugs in the text.
R: It is policy in our center to minimize the use of benzodiazepines and other hypnotic drugs because these drugs have a higher rate of side effects in these patients. Hypnotic drugs were withhold at least 1 week before of the sleep recording studies in a minority patients that occasionally used them .
Q3
The authors explored the following polygraphic sleep measures: AHI, number of O2 desaturation episodes per hour, minimum O2 and average nocturnal O2 saturation. Please, add also the time spent with O2 saturation below 90%.
R: Unfortunately we did not collect this information in our study.
Q4
Figure 2 was irrelevant. Please, delete.
R: We believe that this Figure which face to face compares renal transplant patients at the baseline study with general population data stratified for age, gender and BMI is important to show that after transplantation SDB regresses almost completely. This was the starting point for us following patients over time. We hope that the referee may agree that we maintain this Figure .
Q5
Page 5, line 193: Delete correlations between AHI and age, BMI.
R: We are aware that these correlations seem obvious. The same correlations were further tested in the Generalized Estimating Equation model presented in Table 2 and Supplementary Table 2 and therefore we believe useful to show also the corresponding unadjusted correlations. We hope that the referee may agree in maintaining these crude associations in the text.
Q6
Change the title underling the importance of BMI as predictor of SDB in transplant patients.
R: The new title is “Long Term Changes in Sleep Disordered Breathing in Renal Transplant Patients: Relevance of the BMI”
Q7
Please, include among limitations that more than one third of patients were studied only at baseline and that only 44 patients had three PSG studies.
R: Our statistician notes that the fact that we applied the Generalized Linear Equations approach for data analysis does not make the inclusion of patients with measurements only at baseline or with 2 measurements as a study limitation. However, to comply with the request of the referee we have now mentioned the problem at the start of the study limitations paragraph where we say” One third of patients were studied only at baseline and only 44 patients had three sleep recording studies. However, the Generalized Estimating Equations model adequately deals with the unequal number of patients at various time points and with the variable time interval between successive recordings (see new reference 29).”.
Reviewer 3 Report
The authors investigated both cross-sectionally and longitudinally stability and changes in sleep-disordered breathing among a sample of individuals after renal transplantation. To do so, the authors assessed a quite large sample. Results showed that the AHI increased, or the other way around; sleep-disordered breathing worsened. Importantly, a higher BMI was the strongest predictor of higher AHI scores. Overall, the topic is timely; issues of sleep-disordered breathing a major health problem, but appears to be particularly overlooked in “somatic” patients such as individuals after renal transplantation.
Abstract: Report participants’ mean age and gender ratio.
Avoid/delete p-values, as p-values without further statistical indices are not useful (Wasserstein, Schirm, & Lazar, 2019).
Key-words; avoid abbreviations, or spell-out all abbreviations, when introduced for the first time.
Introduction;
“…. Accordingly, in a case-control study by us the prevalence of SDB among renal transplant patients…”, please find a smoother wording.
“Furthermore, ….. grafting”, this sentence consists of about 62 words spanning over 4.5 lines; please do not overestimate the reader’s working memory; or the other way around: please increase the ease of readability.
“SDB in the dialysis population”: never ever start a sentence with an abbreviation; do not denote a person or a population by their illness; rather: “… in people with dialysis…”
“It is well know (sic!) that risk factors change dramatically after transplantation.”; even, if this is well known, nevertheless, please report references.
“SDB in the …. after kidney grafting.”; please specify and summarize, if SDBs are an issue of the central nervous system or of the peripherical and mechanic structure or both.
“… the long-term evolution of…”; it is not an evolution, but a development or change.
You cannot test an enlargement of a sample, but you can test hypotheses. Please modify and formulate hypotheses. Further, describe in more details, if and to what extent the present data add to the current literature in an important way. Further, once the reader has read the Abstract, she/he is sensitized to the issue of obesity/overweight. Given this, the authors must introduce a chapter dealing with association and the state-of-the-art as regards SDBs and BMI.
Method: “..and WRITTEN informed consent…”; for the sample, we need much more information.
I strongly suggest to introducing a chapter called Study procedure or Study design; this would help the reader to understand the frame of the study.
“….Baseline polysomnographic studies were performed…”; I think you performed polysomnographic assessments, but not studies.
“….when patients had achieved a stable condition at least 4 months after transplantation and were repeated at various time intervals, always while patients were stable and without intercurrent problems, over a median follow up of 35 months (whole range 4 to 110 months; interquartile range 24-46 months).”; this sentence is too complicated; please simplify.
A further subchapter should be: Assessment of SDBs
Statistics: report a reference for the Tweedie procedure; “SPSS for Windows (version 24.0, Chicago, Illinois, USA).”; here, there are several mistakes; the text should read: SPSS® 24.0 (IBM Corporation, Armonk NY, USA) for Windows®.
Results. Table 1; please report all statistical indices, and not only the p-value; as mentioned, the “significance” of a p-values largely depends from the sample size, and not from the means and standard deviations.
Figure 3 needs Figure captions.
Perhaps there was an issue when uploading the files; at any rate the Discussion and Conclusion section are missing.
Overall, while the content of the study is nicely made and above all of clinical and practical importance, the manuscript needs a thorough revision. To get an idea about the structure of a paper, the authors might a close look at the following papers published at MDPI (Jalilian et al., 2020) or at any other paper published at the IJERPH.
References
Jalilian, F., Mirzaei-Alavijeh, M., Ahmadpanah, M., Mostafaei, S., Kargar, M., Pirouzeh, R., . . . Brand, S. (2020). Extension of the Theory of Planned Behavior (TPB) to Predict Patterns of Marijuana Use among Young Iranian Adults. Int J Environ Res Public Health, 17(6). doi:10.3390/ijerph17061981
Wasserstein, R. L., Schirm, A. L., & Lazar, N. A. (2019). Moving to a World Beyond “p < 0.05”. The American Statistician, 73(sup1), 1-19. doi:10.1080/00031305.2019.1583913
Author Response
Referee #3
Q1
Abstract: Report participants’ mean age and gender ratio.
R: we added this information in the abstract
Q2
Avoid/delete p-values, as p-values without further statistical indices are not useful (Wasserstein, Schirm, & Lazar, 2019).
R: We now specifically clarified in the statistical analysis section the tests we applied, including Table 1 data, according to the nature and distribution of the variables being compared among groups (pg 3, L126-128).
Q3
Key-words; avoid abbreviations, or spell-out all abbreviations, when introduced for the first time.
R: We spelled out BMI and CKD in the key words
Q4
Introduction;
“…. Accordingly, in a case-control study by us the prevalence of SDB among renal transplant patients…”, please find a smoother wording.
R: We now say (Pg 2 L47-48) “Accordingly, in a case-control study the prevalence of SDB among renal transplant patients was similar to that in the general population”
Q5
“Furthermore, ….. grafting”, this sentence consists of about 62 words spanning over 4.5 lines; please do not overestimate the reader’s working memory; or the other way around: please increase the ease of readability.
R: We substantially shortened the sentence (Pg2 L48-50) which is now 34 words.
Q6
“SDB in the dialysis population”: never ever start a sentence with an abbreviation; do not denote a person or a population by their illness; rather: “… in people with dialysis…”
R: We rephrased the sentence which now is “In dialysis patients SDB…(pg 1, L42)”
Q7
“It is well know (sic!) that risk factors change dramatically after transplantation.”; even, if this is well known, nevertheless, please report references.
R: We have now dropped “it is well known” and quoted four references (21-23) (pg 2 L51)
Q8
“SDB in the …. after kidney grafting.”; please specify and summarize, if SDBs are an issue of the central nervous system or of the peripherical and mechanic structure or both.
R: We have now improved a sentence at pg 1 L42- pg 2 L46 and say “In dialysis patients SDB is a multifactorial disorder which in part depends on compromised upper airway stability (rostral fluid shift overnight) [8] [9], chronic metabolic acidosis [10] and altered central and peripheral chemosensitivity, and probably on the accumulation of still uncharacterized uremic toxins [11 that reduce airway muscle tone”.
Q9
“… the long-term evolution of…”; it is not an evolution, but a development or change.
R: At Pg 2 L 68 we now say “the long term changes over time in SDB “
Q10
You cannot test an enlargement of a sample, but you can test hypotheses. Please modify and formulate hypotheses. Further, describe in more details, if and to what extent the present data add to the current literature in an important way. Further, once the reader has read the Abstract, she/he is sensitized to the issue of obesity/overweight. Given this, the authors must introduce a chapter dealing with association and the state-of-the-art as regards SDBs and BMI.
R: We rephrased the sentence that is now (Pg2 L65-68) : “In this study we investigated whether the improvement in SDB achieved after transplantation is maintained in the long term. To this scope, we enlarged the previous survey at our center [12] from 163 to 221 transplant patients and performed 404 polygraphic sleep studies over a median follow up of 35 months and herein we describe the long term changes over time in SDB in the same cohort.”
As to BMI, at Pg2 L 59-61 we say” In particular, the role of overweight and obesity, a major risk factor for SDB which frequently develops after renal transplantation [22, has never been tested in a study with longitudinal design in this population”.
Q11
Method: “..and WRITTEN informed consent…”; for the sample, we need much more information.
R: Observational studies not applying drugs do not need formal approval by ethical committees in Italy. This question was also posed by the Assistant Editor of the Journal and we have sent her the official Ethical Guidelines of our region to support the statement above.
Q12
I strongly suggest to introducing a chapter called Study procedure or Study design; this would help the reader to understand the frame of the study.
R:As suggested by the referee, we added two sub-headings “study design” (Pg2 L 73) and “procedures” (Pg2 L82)
Q13
“….Baseline polysomnographic studies were performed…”; I think you performed polysomnographic assessments, but not studies.
R: We reworded the sentence as suggested by the referee (assessments instead of studies)
Q14
“….when patients had achieved a stable condition at least 4 months after transplantation and were repeated at various time intervals, always while patients were stable and without intercurrent problems, over a median follow up of 35 months (whole range 4 to 110 months; interquartile range 24-46 months).”; this sentence is too complicated; please simplify.
R: This information is now given in a shorter sentence under the sub-heading “study design” (Pg2 L74-77) and in a second sentence at the start of “Study Procedures” (Pg 2 L83-84).
Q15
A further subchapter should be: Assessment of SDBs
R: We created a separate sub-heading for SDBs assessment
Q16
Statistics: report a reference for the Tweedie procedure; “SPSS for Windows (version 24.0, Chicago, Illinois, USA).”; here, there are several mistakes; the text should read: SPSS® 24.0 (IBM Corporation, Armonk NY, USA) for Windows®.
R: We now added a reference, Ref 30, for the Tweedie distribution (Tweedie, M. C. K. (1984). “An Index Which Distinguishes between Some Important Exponential Families.” In Statistics: Applications and New Directions - Proceedings of the Indian Statistical Institute Golden Jubilee International Conference, edited by J. K. Ghosh and J. Roy, 579–604. Calcutta: Indian Statistical Institute). We corrected the text along the indication by the referee.
Q17
Results. Table 1; please report all statistical indices, and not only the p-value; as mentioned, the “significance” of a p-values largely depends from the sample size, and not from the means and standard deviations.
R: We now specifically clarified in the statistical analysis section the tests we applied according to the nature and distribution of the variables being compared among groups (pg 3, L126-128).
Q18
Figure 3 needs Figure captions.
R: We added a legend to Figure 3
Q19
Perhaps there was an issue when uploading the files; at any rate the Discussion and Conclusion section are missing.
R: We realized the problem a few days after submission and asked the Journal to upload the complete, final version. I am sorry that the referee received the incomplete version.
Reviewer 4 Report
I reviewed the manuscript entitled "Long Term changes in Sleep Disordered Breathing in Renal Transplant Patient.”
The work is well written and clarifies the results clearly and consciously. The test is well applied and results with the correct statistics.
However the paper presents a major limitation.
1)First the most serious limitation of the study is used methodology .
78 How have you assess that patient is stable ? physical examination? Lab test?
The serious limitation is lack of polisomnography.
In the study respiratory polygraphy was used, and study was conducted not in sleep laboratory. Where was it conduct? A hospital ward?
The patient was not in single room, so it was difficult to avoid sleep disturbance. The room was inspected by nurse, so sleep certainly was interrapted. If sleep was disturbed and there was no EEG (PSG), it is not certain that patient was sleeping , thus AHI can be false.
93 The polygraphy was scored in 30-second periods. Are you sure? 30-s periods is used to assess PSG, not polygraphy.
94 Definition of apnea and hypopnea are incorrect.
The apnea is not defined as complete cessation of flow, but as a reduction in the amplitude of breathing by ≥30% for ≥10 s.
-
Hypopnea should be defined as a reduction in the amplitude of breathing by ≥30% for ≥10 s with a ≥4 % decline in blood oxygen saturation.
99 The classification used in the study comes from a Wisconsin study (1993) and should be not used anymore. The mild >5, moderate >5<15, and severe >30 according to AASM - should be used.
2) Figure 1- there is a mistake -polisomnography. „Polygraphy” should be used.
3) The studied group was 221 patient . However, OSA was diagnosed only in 55 patients. There is no information about prevalence of severe OSA (AHI >30).
210 The mean AHI was very low studied group (AHI-1.8) . On third visit AHI was 3.6- what type of epizodes ? central or obstructive? Most of the subjects had mild OSA (AHI<15), which is not correlated with increased cardiovascular risk. It should be mentioned in limitation of the study.
4) Table 1
„Number of O2 desaturation episodes”- nomenclature is incorrect. „ Oxygen desaturation index (ODI) should be used.
The polygraphic data are incomplete. Apnea index, CAI, OAI, MAI, HI and others are missed.
190 AH? Or AHI
253 These data are unexpected and should be disccused . Probably AHI was underestimated because of disturbed sleep ?
190/191 – central episodes was more freguent than obstructive episodes – why ? it should be discussed. Why AHI correlated with BMI? Central episodes usually did not correlate with body mass index.
194 number of episodes or index? These are different data, be precise.
Figure 3 AHI >15 is not severe, severe is AHI>30
Author Response
Referee#4
Q1
The most serious limitation of the study is used methodology. How have you assess that patient is stable ? physical examination? Lab test?
R: With “clinical stability” we intend without any intercurrent acute clinical problem like kidney rejection episodes or infections or other acute clinical problems and with stable renal function (serum creatinine and GFR) and stable metabolic parameters (glucose and serum electrolytes).
Q2
The serious limitation is lack of polisomnography. In the study respiratory polygraphy was used, and study was conducted not in sleep laboratory. Where was it conduct? A hospital ward? The patient was not in single room, so it was difficult to avoid sleep disturbance. The room was inspected by nurse, so sleep certainly was interrapted. If sleep was disturbed and there was no EEG (PSG), it is not certain that patient was sleeping , thus AHI can be false.
R: We are aware that polygraphy is not the golden standard for the diagnosis of sleep apnea. However, sleep laboratories –including the one at our hospital- are overbooked with studies to be performed for clinical reasons. Performing the large number of sleep studies we have done over the years (n=404) wouldn’t have been possible had we adopted polysomnography. Even though inferior to polysomnography, polygraphy is considered as an acceptable alternative to polysomnography (Eur Respir J 2004; 24: 443–448; Arch Bronconeumol. 2005 Feb;41(2):71-7.; Thorax 2011;66:567e573), which is of logistic importance in large studies like ours.
Q3
The polygraphy was scored in 30-second periods. Are you sure? 30-s periods is used to assess PSG, not polygraphy.
R: We confirm that our polygraphic recorders allow scoring in 30 sec. periods.
Q4
Definition of apnea and hypopnea are incorrect. The apnea is not defined as complete cessation of flow, but as a reduction in the amplitude of breathing by ≥30% for ≥10 s. Hypopnea should be defined as a reduction in the amplitude of breathing by ≥30% for ≥10 s with a ≥4 % decline in blood oxygen saturation.
R: We agree with the referee that today’s criteria are less strict than in the past. However, our study is a long study started in March 2004 and our protocol was established according to standard, internationally accepted criteria of the time (2004). We could only maintain the same criteria throughout the study to ensure analytical coherence. We hope that the referee may agree on this issue .
Q5
The classification used in the study comes from a Wisconsin study (1993) and should be not used anymore. The mild >5, moderate >5<15, and severe >30 according to AASM - should be used.
R: As discussed above, in 2004 –the start of our study- we could only adopt the definition of SDB of that time. For analytical coherence we had to maintain the same definition over time. One additional reason why we considered moderate and severe SDB as a combined category was the small number of patients with severe SDB in our cohort (8/221 at baseline, 9/139 at the first and 4/44 patients at the second visit.
Q6
2) Figure 1- there is a mistake -polisomnography. „Polygraphy” should be used
. R: We have now changed polysomnography into polygraphy
Q7
The studied group was 221 patient . However, OSA was diagnosed only in 55 patients. There is no information about prevalence of severe OSA (AHI >30).
R: The number of patients with severe SDB was small. 8/221 patients had AHI >30 at baseline, 9/139 patients at the first visit and 4/44 patients at the second visit.
Q8
The mean AHI was very low studied group (AHI-1.8) . On third visit AHI was 3.6- what type of epizodes ? central or obstructive? Most of the subjects had mild OSA (AHI<15), which is not correlated with increased cardiovascular risk. It should be mentioned in limitation of the study.
R: We agree that on average the number of AHI episodes was small and even though doubling over time, remained relatively small. As shown in Fig.3 and as discussed at Pg8 L290-295), over a median follow up of 3 years, the prevalence of moderate to severe SDB in the baseline study was just 8% but increased to 13% at the second visit and to 20% at the third visit. In a large study in 2921 elderly people in the general population a 20% prevalence in SDB was seen only in patients in the age range between 65 and 72 years [reference 34]. i.e. in people about 20 years older than renal transplant patients in the present study (average age 47±12 years ). We also discussed that (pg8 L296-299) an AHI between 4 and 12, an alteration we found in 1 out of 4 male and in 1 out of 11 female transplant patients in this study, entails a 75% excess mortality risk of any cause, and stroke in patients with suspected sleep apnea[Ref 37]. Furthermore, a recent thorough analysis aimed at optimizing the thresholds for mild, moderate and severe SDB (J Sleep Res. 2019;28:e12855) on the basis of the risk of death remarked that the separation among these categories for the risk of death is lower than the present thresholds (>5, > 15 and >30) being >3, > 9 and >24. This analysis also emphasizes that progressively more severe degrees of SDB entail a parallel progressive increase in the risk of death .
Q8
Table 1 Number of O2 desaturation episodes”- nomenclature is incorrect. „ Oxygen desaturation index (ODI) should be used.
R: We have now corrected this inappropriate nomenclature
Q9
The polygraphic data are incomplete. Apnea index, CAI, OAI, MAI, HI and others are missed.
AH? Or AHI
R: As we specified in the reply to Q4 by Referee#1 and Q#1 to Referee 2, in rechecking data about the type of apnea we realized that the figures about central and mixed apnea at baseline were wrong. Ours is a study started in 2004 and information on apnea types was not systematically transcribed in the study data base. We planned to recheck all polygraphic recordings of our study and asked to the Journal an extension of the deadline for resubmission. Unfortunately we discovered that recordings from March 2004 to August 2007 (our first survey including 163 patients which we extended to 221 to form the basis of the present longitudinal study) were irretrievable. We are aware that this information is of relevance. Even though restricted to the AHI, we feel that our study contains valid, novel data that may contribute to knowledge in the field. We are pretty sure that the vast majority of apneas were obstructive but we cannot provide data to support this statement. If the referees believe that information about apnea types is obligatory, we are sorry but we can only withdraw our submission.
Q10
These data are unexpected and should be disccused . Probably AHI was underestimated because of disturbed sleep ?
R: We cannot precisely reply to this question because we did not perform the EEG in our study. However, the nurses that attended polygraphic studies did not report particular sleep problems in the majority of recording sessions.
Q11
central episodes was more freguent than obstructive episodes – why ? it should be discussed. Why AHI correlated with BMI? Central episodes usually did not correlate with body mass index.
R: See rely to Q9
Q12
number of episodes or index? These are different data, be precise.
R: I am unclear at which part of the text this note refers. To the AHI? In the methods we say that this index is the number of apnea and hypopnea events per hour, which we believe is correct.
Q13
Figure 3 AHI >15 is not severe, severe is AHI>30
R: We corrected this mistake
Round 2
Reviewer 1 Report
The authors addressed my comments.
However, please add information to the methods section of the manuscript regarding the lack of diffraction of central and obstructive events in part of the study group. This should also be discussed as a limitation of the study.
In supplemental Table 1:
the group description should be as follows: AHI<5, 5≤AHI<15, AHI>15, it would be more clear
either keep the % in the brackets next to the parameter or next to the number, not both
Author Response
However, please add information to the methods section of the manuscript regarding the lack of diffraction of central and obstructive events in part of the study group. This should also be discussed as a limitation of the study.
R: At Pg9 L319-320 we now frankly say” Furthermore, for technical reasons we could not produce an analysis according to apnea type and the time spent with SaO2 < 90%”.
In supplemental Table 1: the group description should be as follows: AHI<5, 5≤AHI<15, AHI>15, it would be more clear either keep the % in the brackets next to the parameter or next to the number, not both
R: In Table 1 we corrected the wrong description (> <) according to the suggestion by the referee and eliminated (%) in the first column
Reviewer 2 Report
The authors should include among the limitations of the study that information on (1) the type of apneic events, and (2) the time spent with O2 saturation below 90% are lacking.
Author Response
The authors should include among the limitations of the study that information on (1) the type of apneic events, and (2) the time spent with O2 saturation below 90% are lacking
R: At Pg9 L319-320 we now frankly say” Furthermore, for technical reasons we could not produce an analysis according to apnea type and the time spent with SaO2 < 90%”.
Reviewer 3 Report
Overall, the quality of the manuscript improved dramatically, and in my opinion, the authors are on the final stretch close to the arrival.
There are some further bugs to fix.
For the statistics, in Table 1 and Table 1S, it is not sufficient to mention “ (see statistical methods)”; if as a reader (and above all as a Reviewer), I get the impression that I have to work harder than the authors, then, basically, there is something wrong. So, please specify the statistical procedures and indices directly where the reader needs these information. In this regard, you’ll find excellent examples in (Ahmadpanah et al., 2019; Mirmosayyeb et al., 2020); while the content of these publications do not touch the content of the present manuscript, both publications were published at the same publisher, that is MDPI.
Smaller points:
288 “about 20years”; insert space
299 “400polygraphic”; insert space
288-291; “The recurrence….sleep apnea[37].”; insert space between apnea and the squared bracket; next; the sentence is 3 lines long; do not overestimate the reader’s and reviewer’s working memory”
Tables; never use vertical bars.
Overall; well done and in bocca al lupo for the final round, and in my opinion, the content of the present manuscript is a real add-on to the current literature.
References
Ahmadpanah, M., Arji, M., Arji, J., Haghighi, M., Jahangard, L., Sadeghi Bahmani, D., & Brand, S. (2019). Sociocultural Attitudes towards Appearance, Self-Esteem and Symptoms of Body-Dysmorphic Disorders among Young Adults. Int J Environ Res Public Health, 16(21). doi:10.3390/ijerph16214236
Mirmosayyeb, O., Brand, S., Barzegar, M., Afshari-Safavi, A., Nehzat, N., Shaygannejad, V., & Sadeghi Bahmani, D. (2020). Clinical Characteristics and Disability Progression of Early- and Late-Onset Multiple Sclerosis Compared to Adult-Onset Multiple Sclerosis. J Clin Med, 9(5). doi:10.3390/jcm9051326
Author Response
For the statistics, in Table 1 and Table 1S, it is not sufficient to mention “ (see statistical methods)”; if as a reader (and above all as a Reviewer), I get the impression that I have to work harder than the authors, then, basically, there is something wrong. So, please specify the statistical procedures and indices directly where the reader needs these information. In this regard, you’ll find excellent examples in (Ahmadpanah et al., 2019; Mirmosayyeb et al., 2020); while the content of these publications do not touch the content of the present manuscript, both publications were published at the same publisher, that is MDPI.
R: We have now added the specific tests applied in the legend to Table 1 and 1S (the tests are described in the legend and identified with * ^ # in the “P” column)
Smaller points:
R:We made all mistyping corrections identified by the referee
288 “about 20years”; insert space
299 “400polygraphic”; insert space
288-291; “The recurrence….sleep apnea[37].”; insert space between apnea and the squared bracket; next; the sentence is 3 lines long; do not overestimate the reader’s and reviewer’s working memory”
Tables; never use vertical bars.
Reviewer 4 Report
Thank you for your aswers and changes.
The lack respiratory parameters ( AI, OAI, CAI MAI, HI) should be inserted in „limitations” and I accept the paper.
Author Response
Thank you for your answers and changes.
The lack respiratory parameters ( AI, OAI, CAI MAI, HI) should be inserted in „limitations” and I accept the paper.
R: At Pg9 L319-320 we now frankly say” Furthermore, for technical reasons we could not produce an analysis according to apnea type and the time spent with SaO2 < 90%”.